# Invited perspectives: An insurer's perspective on the knowns and unknowns in natural hazard risk modelling

Madeleine-Sophie Déroche[1]

[1]AXA Group Risk Management, Paris, 75008, France

**Correspondence:** Madeleine-Sophie Déroche (madeleinesophie.deroche@axa.com)

**Abstract.** This paper analyses how the current loss modelling framework that was developed in the 1990's to respond to hurricane Andrew market crisis falls short in dealing with today's complexity. In effect, beyond reflecting and supporting the current understanding and knowledge of risks, data and models are used in the assessment of situations that have not been experienced yet. To address this question, we considered the (re)insurance market's current body of knowledge on natural hazard loss modelling, the fruit of over 30 years' research conducted by (re)insurers, brokers, modelling firms, and other private companies and academics in the atmospheric sciences, geosciences, civil engineering studies, and data sciences among others. Our study shows that to successfully manage the complexity of the interactions between natural elements and the customer ecosystem, it is essential that both private companies in the insurance sector and academia continue working together to co-build and share a common data collection and modelling. This paper (i) proves the need to conduct an in-depth review of the existing loss modelling framework and (ii) makes it clear that only a transdisciplinary effort will be up to the challenge of building global loss models. These two factors are essential to capture the interactions and increasing complexity of the three risk drivers – exposure, hazard, and vulnerability – thus enabling insurers to anticipate and be equipped to face the far-ranging impacts of climate change and other natural events.

## 1 Introduction

The mission of Property and Casualty (P&C) insurers is to effectively protect clients' property and activities while ensuring the solvency of the company. Though insurers develop ever-increasing products to respond to clients' specific needs, P&C insurance in essence consists of two segments, the (i) retail business for home and car owners and (ii) commercial business for corporate clients. Insurance protection goes beyond risk transfer (i.e., the payment of a premium against future claims); it also encompasses prevention actions such as reinforcing customers' risk awareness and proposing adapted protective solutions. For example, in commercial business, technical risk experts perform on-site visits to evaluate the state of buildings and identify potential vulnerabilities to natural hazards. The objective is to assess how natural hazards could generate damage either to the buildings themselves (e.g. storage warehouses, data centers, shopping centers) or to their contents (e.g. machinery, production chains, stock), and if such damage could cause business interruption (e.g. employees / clients / suppliers being unable to access the building for N days resulting in a loss of turnover or profits). Prevention measures like elevating goods or machinery in the event of flooding are then suggested or imposed post-assessment to reduce the risk and adjust the premium.

In the retail business with its mass of clients, protection actions have to be taken globally instead of individually. For example, after the Great Fire of London in 1666 that destroyed most of the city's buildings, made of wood at that time, insurance premium rates were lowered for buildings made of brick to encourage brick constructions instead of wood thus reducing the fire risk in London.

To achieve their mission of protection, it is essential for insurers to identify and quantify the risks associated with the underwritten policies. All along the P&C insurance value chain, a vast range of data feed an equally vast range of models to estimate the losses for the varying probabilities and magnitudes of all the underwritten risks, be they natural hazards, financial, or cyber. These models serve to support decision-making from the actual underwriting and pricing of an individual or corporate's policy to the setting and optimization of the reinsurance programs at the insurance company level.

The regulatory environment also plays a significant role in validating the models (re)insurers use to assess risks and ultimately, better protect the end-customer. Regulations require (re)insurers to notably assess the extreme losses of all their risks to determine their minimum level of economic capital to ensure the (re)insurance companies' solvency in the event of intensely severe years. The European Solvency 2 regulation is a case in point. (Re)insurers with Europe-based headquarters are required to annually project their losses for a 200-year return period shock along with the associated risk management actions such as

the purchasing of reinsurance covers. This estimated amount of loss determines the level of capital (re)insurers have to bear in their owned funds to resist such a shock if it were to occur in the following year. The models used to assess this loss require approval and any change is thoroughly monitored by regulatory authorities.

Sound and adaptive risk assessment and management are built over time through a continuous reassessment of insurers' understanding of: the "known knowns", what we know we know; the "known unknowns", what we know we do not know; and

the "unknown unknowns", what we don't even know we do not know (Girard, 2009). This reassessment process induces a knowledge cycle: data continuously supports the current understanding and knowledge of a risk, that is, what we know we know. On the basis of this understanding, models are also built to support assessing situations that have yet to be experienced such as extreme events, that is, what we know we don't know. However, the occurrence of unknown unknowns triggered by natural and organizational issues that insurers either ignore or have yet to understand, points to a pressing need to upgrade data

collection, modelling methods and tools to perpetually enhance the view of risk and further insights for the decision-making process.

Prior to focusing on the scientific and technical advances made to keep refining what we know about the risk drivers, exposure, hazard, and vulnerability, and how to increase insurers' preparedness for the unknown, it is important to recall how the reassessment process engendered the natural hazard loss modeling framework from the outset.

## 2   Natural hazard modelling: a brief overview

### 2.1   The co-influence of (re)insurance market and natural hazard modelling

The actual assessing of natural event related costs has greatly evolved over the past 30 years. At first, so-called catastrophe models focused on the modelling of extreme losses to assess the risk of a portfolio (i.e. large ensembles of insured buildings).

Before the 1990s, catastrophe modelling consisted in extrapolating the loss experience to estimate extreme losses. This loss experience was usually limited and recorded as a total amount of loss per event and per insurer or per event and for the whole (re)insurance market. The data were thus too coarse to capture the three risk drivers' individual impact on the losses: the exposure (e.g. what if the exposure is located in a more / less risky area), the hazard itself (e.g. what portion of the losses are generated by a storm surge versus wind in the case of a tropical cyclone), the vulnerability (e.g. how effective the flood defenses, building codes are). As a result, while the data and the resulting modelling failed to take into account individual effects when assessing extreme losses, it did reflect the state of what was known by insurers and public authorities at that time. Hurricane Andrew in 1992 and its unexpected impact was a game changer for modelling natural hazard-generated losses (Grossi et al., 2005; Mitchell-Wallace, 2017). According to McChristian (2012), before Hurricane Andrew, the loss assessment for an event of that strength was $4 to $5bn. This is three times lower than Hurricane Andrew's actual loss at $15 bn. Insurers underestimated their exposure as well as their exposure's vulnerability to such an event. McChristian (2012) also indicates that though past experienced losses were adjusted to reflect current macro-economic trends, they failed to capture the increasing population over coastal areas. In the aftermath of Hurricane Andrew, a collective realization grew for the need to both separately characterize the three drivers of the risk - exposure, hazard and vulnerability - and model their interconnections. Catastrophe modelling therefore evolved from a statistical extrapolation to a framework divided into 4 components as shown in Figure 1; one component by risk driver (exposure, hazard and vulnerability), and one component that contains the insurance policies' financial conditions and its modelling.

The occurrence of natural disasters, in particular those with a strong impact for the (re)insurance market, continues to feed research insofar that the research is in turn integrated into the hazard and vulnerability components of the loss modelling framework every 2 to 5 years. This is how, the successive 1999 occurrence of the two extreme European winter windstorms Lothar and Martin triggered the introduction of the serial clustering effect in modelling the frequency of European winter windstorms (Mitchell-Wallace, 2017). The serial clustering effect refers to the higher probability that two extreme winter windstorms occur in a short period of time, under particular atmospheric conditions (Vitolo et al., 2009; Pinto et al., 2013; Priestley et al., 2017). Prior to these windstorms, the assumption used to calculate the occurrence probability of European winter windstorms followed the Poisson distribution and thus failed to allow for the increased probability of successive events. As shown by Priestley et al. (2018), the clustering effect has a significant impact on the estimation of yearly aggregated losses and therefore on the sizing and the wording of reinsurance covers.

Within the reinsurance market, the use of catastrophe models - developed internally or licensed through third-party vendors - has grown in the aftermath of Hurricane Andrew. For insurers to cede their risk they must provide their exposure information to reinsurers so they can conduct a loss assessment prior to estimating the reinsurance premium corresponding to the accepted risk. Today, catastrophe models continue to be used primarily to set reinsurance programs (i.e. total capacity and pricing).

The implementation of regulation has prompted insurers to use catastrophe models, mainly licensed by third-party vendors, as tools to assess the risk, define the risk appetite, and set the solvency capital requirement. For example, the Solvency 2 regulation implemented in Europe in 2016 requires (re)insurers with Europe-based headquarters to annually assess their loss for a 200-year return period shock. (Re)Insurers conduct this assessment for all the risks they are exposed to. They then aggregate these

estimated losses to determine the total potential loss and the economic capital they have to bear in their owned funds. To achieve this assessment, (re)insurers have two options: either to use the so-called Standard Formula, calibrated on market exposure and at a relatively coarse granularity, or to develop an internal view of their risk that requires regulator approval. Most (re)insurers choosing to develop their own view of natural hazard risk use one or several models licensed to third-party vendors; others develop their own suite of models. Model evaluation becomes a necessary activity for assessing the model's strengths and limitations leads to gaining in understanding and in taking ownership of the model. When (re)insurers opt for using third-party models, adjustments may be defined and applied to the models' loss estimation to address identified limitations (e.g. a non-modeled peril such as storm surge induced by windstorms). (Re)insurance companies also invest in the development of in-house models either on scopes where no third-party vendors model is available or to gain in flexibility and transparency.

In the past few years, both the scope and use of catastrophe models have evolved. Indeed, to estimate the insurance premium of an average risk, insurers are now as interested in capturing small frequent events as large rare ones. In the hazard module, the full spectrum of events (i.e. moderate/intense; frequent/rare events) is considered. In the vulnerability module, vulnerability curves cover the entire range of hazard intensity. As for the modelling scope, catastrophe models also exist for man-made perils such as cyber and terrorism. To reflect these evolutions, we will use the term 'natural hazard models / modelling' as it allows for greater precision on the model's targeted scope and reaffirms the use of these models for other purposes than the analysis of extreme events.

## 2.2   Natural hazard modelling framework

The loss modelling framework is composed of 4 components, namely exposure, hazard, vulnerability and financial components (Figure 1). The description below provides a brief introduction. Greater details on the different components can be found in Mitchell-Wallace (2017).

The exposure component contains the insurance portfolio's information: the buildings' location, and their key physical properties (e.g. structure, occupancy, year of construction...). The hazard component contains a synthetic catalogue of several tens of thousands of events that represent the range of possible and plausible events for a given natural hazard (e.g., Asia typhoon, US ground shaking, Europe severe convective storms, ranging from small frequent events to extreme rare events. Each event is characterized by a footprint (i.e., the maximum intensity over event duration) and an annual occurrence probability. The vulnerability component is composed of vulnerability curves that translate the hazard's intensity into a building damage ratio. Ideally, there is one vulnerability curve for every combination of a building's physical properties. Finally, the financial component contains the insurance contract's financial data: the sum insured corresponding to the coverage (building, content, business interruption), the deductibles and limits, as well as the coinsurance programs or reinsurance treaties, if any.

For every event of the hazard component and for every building in the insurance portfolio, the 3-step loss modelling process consists in:

1. Intersecting the building's location with the event footprint to obtain the location's hazard intensity value.

2. Factoring in the hazard intensity value and the building's physical properties and using the vulnerability curve reflecting the building's characteristics to derive the corresponding damage ratio.

3. Applying the damage ratio to the insured value of the building, as specified in the financial module, to provide a loss amount prior applying the financial conditions to the loss amount to get the ultimate loss borne by the insurance company.

130 The primary outputs of natural hazard models are exceedance probability distributions representing the probability to exceed a certain amount of loss. Two distributions are commonly used: the one for the annual maximum loss - Occurrence Exceedance Probability (OEP)-, and the one for the annual aggregated loss – Aggregate Exceedance Probability (AEP). The Annual Average Loss (AAL) is also frequently used for budget planning for instance. Analyses of building losses are aggregated at granularities going from the building level to the portfolio level to characterize the probability to exceed an amount of loss.

135 This granularity is set in function of an analysis' objective, i.e. policy underwriting or portfolio management.

The loss modelling process is supported by a platform that contains (i) the data of each component stored in a specific format (e.g. csv or netcdf file, digital precision...) and (ii) the code functions that process data and estimate the losses. Until the early 2010s, the loss modelling process could only be performed on proprietary platforms. Launched in 2010, the OASIS[1] initiative's ambition is to provide an open-source loss modelling platform to further transparency and to expand the use of natural hazard

140 modelling beyond the (re)insurance market.

From a business perspective, integrating such a process in daily operational activities requires the run time to take no more than a few hours. As an example of volumes at stake, assuming we have a catalogue of around 30,000 events and a portfolio of 5 million buildings, a total 150 billion rows would be needed to capture and store the risk distribution. This is without taking into account more advanced modelling of randomness and local effect scenarios, that would increase the dimension of outputs

145 by several orders of magnitude. To keep to the expected run time and given the constrained IT infrastructure with limited storage space and a memory limit, the loss modelling platform is to be rationalized and optimized, even if it results in a drop of formatting flexibility and data precision within the 4-components.

Today's IT computation constraints make it necessary to downgrade the quality and sophistication of the researchers' modelling to obtain results within an acceptable period. This compromises the assessment that could be attained and engenders a

150 precision gap between what research produces and the derivative data ultimately integrated in the loss modelling framework. For example, the severity of natural events is captured in the hazard component through the use of hazard footprints defined as the maximum hazard value (e.g., windspeed, flood depth, peak ground acceleration) at each grid cell of the considered area over the duration of the event. The information relative to the event's duration and to the hazard value's evolution over time however are lost, even though both of these parameters affect the damage assessment of a building.

The 4-component loss modelling framework makes it easier to identify the areas where, component by component, a more in-depth investigation is needed to refine data collection and modelling. The next section focuses on three of the loss modelling framework's components highlighting where (i) a thorough and systematic data collection needs to be put in place, and (ii) the loss modelling framework requires investment to upgrade it and tailor it to respond to insurers' business needs.

---

[1]https://oasislmf.org/our-modelling-platform

# 3 Current challenges in modelling natural hazards

The (re)insurance market's current body of knowledge on natural hazards loss modelling results from over 30 years' research involving private companies like (re)insurers, brokers, and modelling firms and academic researchers in atmospheric sciences, geosciences, civil engineering studies, and data sciences, to name but a few disciplines (Ward et al., 2020). The learning curve has been steep, closely linked to the increase of computer power (e.g., enabling the development and implementation of millions of possible climatic or seismic scenarios) and the collection of increasingly granular observational data (e.g. hazard,

claims, geocoded exposure).

## 3.1 Exposure component

Through an increasing use of natural hazards models, insurers have realized that both data quality and its completeness reduce the uncertainty in the modelling. Over the past 5 to 10 years, insurers have significantly improved the collection process of information characterizing their exposure, namely the coordinates of the location of the buildings as well as the buildings'

physical properties. As mentioned previously, exposure data in the loss modelling process is used (i) to estimate the hazard's severity at the location of the building and (ii) to select the suitable damage curve. The more precise the exposure data, the more accurate the loss evaluation will be. However, as some elements are particularly difficult to get at the time of underwriting individual insurance, the systematic extraction and completion of the data remains a challenge and any missing information needs to be completed once the policy is underwritten either in the exposure database or at a later stage in the modelling.

The increasing volume and precision of geographical information captured by satellites allowed for the development of performant geocoding tools supporting the completion of the exposure database. With an address, it is possible to get the geolocation, the structure of the building, number of floors and even the roof type, all critical drivers of damage for different perils (Ehrlich and Tenerelli, 2013; Castagno and Atkins, 2018; Kang et al., 2018; Schorlemmer et al., 2020). This progress in characterizing buildings' properties along with geolocations, was a major advancement enabling insurers to visualize and analyze their accu-

mulation to natural hazard risk.

When critical information is missing in the exposure database, assumptions are made by using either other data sources to complete the exposure database (e.g. exposure disaggregation to fill in buildings' geolocation) or generic vulnerability curves defined as the weighted average of specific vulnerability curves in the loss modelling process. Any omission on the properties of a building's construction induces an uncertainty on that given building(s)'s exposure that can be quantified through sen-

185 sitivity tests that assess varying combinations of a building's construction properties and the resulting impact on losses. The impact of inferring geolocation might however be greater, depending on the peril in question, as for flood and severe convective storms. Testing the impact on losses of a disaggregation scheme requires running the model using several versions of disaggregated portfolios, which is inconceivable today notably because of run time constraints. The disaggregation technique could also provide a solution to modelling the impact of natural hazard on movable exposure. Today, motor and marine exposures are

190 modelled like buildings'. The geolocation used is the car owner's address or the vessel's home port as specified in the policy

contract. Disaggregating the motor or marine exposure multiple times would give different vehicle locations and hence capture a range of potential losses.

## 3.2 Hazard component

An ever-growing amount of data on the hazard component has been made accessible, refined, and maintained. A multitude of types of data, from observations to model simulations or a mixture of both, substantially support the development of hazard catalogues and their validation. Hazard modelling sets out to characterize, via a hazard events catalogue, the full spectrum of severity and frequency of hazards on a specific geographical area. A review of hazard modelling approaches by peril can be found in Ward et al. (2020). Beyond the perpetual enhancement necessary to complete and refine the view of the risk and to adapt to an ever-evolving environment, uncertainties persist in being only partially quantified due to (i) IT constraints and (ii) the information loss perpetuated by simplifying assumptions to derive data compiled in the loss modelling framework. Resolving these two sources of uncertainties would enable insurers to heighten their understanding of the risk and make sounder business decisions.

Uncertainties in the hazard component come from the input data and the modelling parameters used to generate the stochastic event catalogue. For example, Kaczmarska et al. (2018) quantify how in changing flooding parameters the loss estimates are impacted. Winter et al. (2018) go a step further notably in identifying and quantifying uncertainties present in the production of the hazard events catalogue. Such an analysis first requires running the production of the hazard catalogue several times to test different sets of parameters and secondly running the loss simulation engine multiple times. Including the quantification of uncertainties is costly both in terms of computer power and runtime but should be systematized as a modelling best practice.

As mentioned in Section 2.2, the information relative to the event's duration and to the hazard value's evolution however are lost when generating the events' footprint, i.e. the maximum value of hazard intensity over the duration of the event. In (re)insurance policies, an event's duration is a metric used, within the hours clause, to specify that the (re)insurer will cover all the financial losses accumulated in a defined number of hours, varying depending on the peril. If financial losses are still recorded surpassing the defined number of hours, it will be counted as a second and separate event and activate a double reimbursement from the reinsurer. According to how the reinsurance program is defined, the insurer may have to pay additional fees to get a cover for the second event. Analyzing the impact of the hours clause on the final loss would therefore be beneficial for the (re)insurance market. The loss modelling framework must evolve to allow for more flexibility and more completeness.

## 3.3 Vulnerability component

When a natural event occurs, damage results from the rupture of one or several of the building's components, the level of the rupture depending on the hazard's severity and the components' vulnerability. In the aftermath of the event, reconstruction costs are assessed based on the current material prices and labor costs. However, in post-disaster situations, reconstruction costs may be significantly higher due to a post-event demand surge and inflation. This effect is called post-loss amplification (PLA) and is modelled using a sigmoid function whose calibration remains difficult as (i) it has been observed subsequent to very extreme events and (ii) reconstruction costs or claims available in the historical record includes the PLA effect. As the

PLA may have a substantial impact on the ultimate amount of loss paid by insurers, further research is needed to analyze and model this effect.

Systematic data collection of damage information and its associated hazard magnitude is therefore vital to characterize the impact of natural hazards on buildings and to improve the calibrations not only of the buildings' destruction rate but also of the reconstruction costs in the vulnerability component. New technologies such as drones and satellites provide alternative ways to access impacted areas to collect detailed and granular measurements within a few hours or days of an event's occurrence (Chesnel et al., 2007; Kakooei and Baleghi, 2017). While there has been a substantial increase in the availability of observational data over the past two decades (Yu et al., 2018), further investments should be made to systematically collect: (i) the event's level of hazard severity at the building's location (i.e. values of the relevant hazards' variables leading to the building's damage), (ii) the building's level of damage and the prevention measures if any (concurrently recording all relevant information on the building itself) and (iii) the level of associated repair costs (including information on loss adjustments and economic metrics such as post event inflation). This data collection effort should be a joint public and private sector undertaking to build up a core common knowledge.

A point of attention is the need for data collectors to coordinate and use the same damage scale to avoid duplicating and overlapping datasets that are incomparable. Research initiatives dedicated to gathering various data sources already exist at country level. One such example is the HOWAS database for flood damage in Germany (Kreibich et al., 2017; Kellermann et al., 2020). Could this type of work be extended to the whole of Europe or even more globally? The PERILS[1] initiative is worth mentioning as it is an example of the (re)insurance market's claims data collection initiative. When an event's loss estimation exceeds a defined threshold, the PERILS organization collects claims from the (re)insurers taking part in the consortium. While this data is aggregated at CRESTA[2] level, the initial estimates of the loss ratios are fundamental to establish market's loss benchmarks and derive market vulnerability curves for instance.

While the challenges set out in this section indicate how to improve what we know we don't know, they also highlight the potential limitations of the current loss modelling framework and its simulation platform. The shortcomings of the current loss modelling framework herein described point to the need for an in-depth review of the framework to improve and increase insurers' understanding of natural hazard risk particularly in an ever more connected environment that is described in the next section. From an insurer's perspective, in a context of growing focus on natural hazard impacts, data collection, modelling flexibility, and transparency have become core strategic elements to enhance and gain confidence in its assessment of natural hazard risk. To achieve modelling flexibility and transparency the loss modelling framework will require in-depth changes to absorb the high amount of data and to incorporate uncertainty quantifications. If tackled collectively, data collection, especially relating to damages and claims, could contribute to better city planning and more effective prevention measures, that would in turn increase society resilience.

---

[1]https://www.perils.org/
[2]https://www.cresta.org/

## 4 Future challenges and further needs

Since the building of the loss modelling framework in the 1990s, clients have become more interconnected (Gereffi et al., 2001), and the correlations between natural hazards and regions have also become better understood and quantified (Steptoe et al., 2018; Zscheischler et al., 2020; Tilloy et al., 2020). This section explores 3 elements that would advance natural risk assessment and would support insurers' in their ambition to more accurately project and plan out their business activities insofar as natural hazards.

### 4.1 Introducing a fifth component to quantify uncertainty

As stated in previous sections, the assessment of uncertainty all along the modelling chain constitutes the loss modelling framework's notable shortcoming and the one that requires further investigation. To a certain extent, uncertainty is inherent to modelling and is partly captured in the loss modelling framework today through (i) the primary uncertainty, that is the assumptions and the simulation of the hazard catalogue, and (ii) the secondary uncertainty, that is the damage and loss assessment.

A prerequisite in understanding the uncertainties embedded in the modelling process is comparing and evaluating the models themselves. To date, models are however insufficiently transparent to perform such a comparison. This points to a want for more transparency. In parallel, to move forward, it is fundamental to systematically quantify these uncertainties to change both how we communicate on them and how we manage them. This will enable insurers to take ownership of uncertainties' management and provide them with a tool to ensure on-going model enhancements (Thompson and Warmink, 2016; Doyle et al., 2019).

Incorporating the quantification of uncertainties in the loss modelling framework does make it more costly in terms of computer power and runtime. In light of the rapid evolution of IT, computer power and run time should not be an issue for long. The question will then be how to implement a comprehensive uncertainty quantification scheme. While Beven et al. (2018) suggest a framework to deal with epistemic uncertainty in natural hazard modelling, recent work like (Noacco et al., 2019; KC et al., 2020) has been carried out to address quantifying uncertainty with appropriate methods and tools. Could we not introduce a specific "uncertainty component" that, combining the multiple datasets from the different components, would deal with an ensemble of models and propagate the quantification all along the loss modelling process?

### 4.2 Supply chain modelling

With globalization, clients around the world have become increasingly interconnected and dependent on each other within so-called Global Value Chains (Gereffi et al., 2001; Baldwin and Lopez-Gonzalez, 2015; Phillips, 2018). This dependency became apparent with the 2011 floods in Thailand when Thailand's brutal interruption of microprocessor production led to a halt in global production, a global shortage of microprocessors, and consequently, a loss in benefits for companies producing chips, hard disc drives and other electronic devices (Chopra and Sodhi, 2014; Haraguchi and Lall, 2015).

From an insurer's perspective, suppliers' defaulting in their deliveries due to the occurrence of a natural hazard is not insur-

able, as it is not quantifiable with the current modelling that fails to capture this connection between suppliers and their client producers. Supply chain data has improved (Tiwari et al., 2018; Beorchia and Crook, 2020) and needs to be analyzed further and incorporated into natural hazard loss modelling. This could provide a source of opportunities for insurers to deliver new services to customers while continuing to contribute to advancing research in visualizing and measuring the levels of complexity (volume, direction and intensity of interconnections).

The interconnections between hazards or between clients have yet to be captured even in the latest loss modelling framework. It remains siloed by hazard and region and omits supply chain information. Failing to integrate these interactions may result in instilling a bias in our understanding of the underlying risk. A deeper review of the loss modelling framework is to be conducted to reflect on this new and complex reality.

## 4.3 Forward-looking scenario: modelling the future of natural hazard risks

Natural hazard models have been primarily developed to overcome the limited historical loss record and to assess extreme losses driven by exposure, hazard and vulnerability in the present. They are now envisaged as tools to assess the future of natural hazard risks, in particular in the context of climate change.

To perform this analysis, insurers not only need to project the plausible future scenarios of hazard events (e.g. in the case of climate change impact studies, information provided by climate model simulations) but also to project the evolution of exposure and vulnerability. In this context, the two most pressing questions global insurers need to respond to are: how to gather future projections of population growth or decline and/or wealth worldwide? How will building codes evolve?

Cremen et al. (2022) perform a thorough review of the available literature and provide initial answers to these questions. Such a review is particularly enlightening to enhance the simple initial assumptions that were made, especially for exposure growth and vulnerability. Furthermore, as vulnerability is a crucial element in adapting to climate change impacts, further investigations on the implementation of prevention measures and the quantification of the resulting risk reduction are needed. Finally, while this forward-looking analysis is necessary, its outcomes should be taken with great caution. As Fiedler et al. (2021) highlight, uncertainty around future exposure, hazard or vulnerability projections, is significant and compounds the uncertainty already present in the loss modelling framework.

## 5 Conclusions

To date, models have evolved through the incorporation of new information, without ever undergoing an in-depth transformation. Modifications have stemmed from the observation of the growing number of interconnections – and mutual impacts – at multiple levels: between insured customers and their suppliers and interactions and cross impacts between the disasters-causing natural phenomena. Though this make-do approach has served, it no longer suffices. In today's world where complex intrinsic interconnections exist between natural hazards, exposure, and vulnerability, models fail to fully reflect this reality. They are in want of an in-depth transformation. Only then will they convey and advance the new level of understanding insurers need to cultivate and enable the design and testing of new products and protection mechanisms.

As said in Baum (2015), "threats are rarely completely unknown or unquantifiable". Sometimes what we do not know is already
present in the data or the model but it has yet to be understood or analyzed. We propose reflecting on how to bring together a
transdisciplinary research team composed not only of IT, data science, and the geosciences, but also of civil engineering, urban
planning sciences, and socio-economic sciences to investigate the opportunities to build global loss models for natural hazards
that would deal with the complexity of the interactions of both natural elements and the customer ecosystem. This would enable
insurers to better anticipate the needs of their customers while being better equipped to cope not only with uncertainty but also
the unknown.

*Acknowledgements.* I greatly acknowledge the three anonymous reviewers and the editor (David Peres) for their insightful revisions and
editing. I would like to thank Beatrice Wing for her thorough proofreading of the paper.

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

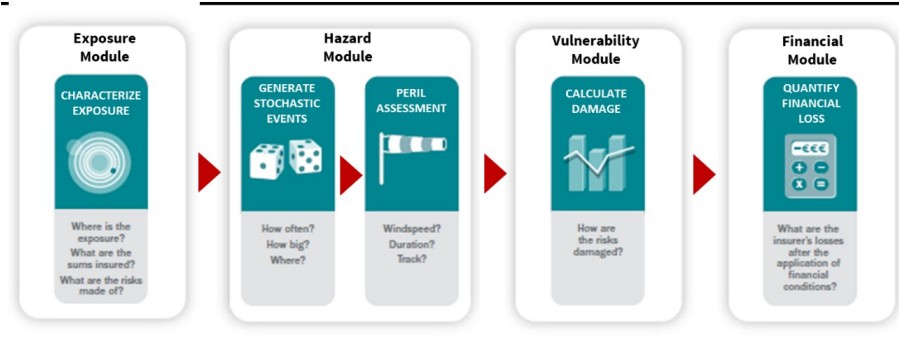

**Figure 1.** Loss Modelling Framework composed of 4 components. A simulation engine is used to intersect the exposure information with the catalogue of hazard events and apply the damage ratio characterized with the vulnerability curve, function of hazard and building characteristics. This operation leads to a loss, gross of any financial insurance conditions. Their application is performed in the financial module.