# Peer review of "Invited perspectives: An insurer's perspective on the knowns and unknowns in natural hazard risk modelling"

_Natural Hazards and Earth System Sciences, 2022_

## Referee Comment (RC3)

**Review „Invited perspectives: Current challenges to face knowns and unknowns in natural hazard risk management - an insurer perspective" by Madeleine-Sophie Déroche (NHESSD)**

The manuscript describes the development of risk models used in the insurance industry over the past three decades and discusses the need for further improvements. The paper touches on a wide range of topics, such as the need for detailed loss and exposure data or the challenge of climate change. A review paper on these topics from the perspective of the insurance industry is very valuable and is desired by the scientific community. However, the paper remains very superficial and lacks a real insurance perspective. I suggest more rigorous structural leasing to better present the content. I strongly encourage the author to rewrite and restructure the manuscript; it is truly of great interest and could have a high impact for the community.

**Major revision points:**

1) I suggest revising the structure of the manuscript. The current version sometimes jumps from one topic to another and sometimes back again (e.g., paragraph 65-87); some statements refer only to individual hazards (e.g., flooding), while the next sentence is a general statement.
   My suggestion is the following structure: (i) Loss / risk model development from a historic perspective, including a detailed discussion of the three components (hazard, exposure, vulnerability); (ii) Cncertainty inherent in each of the components (e.g., uncertainty in hazard modeling due to a lack of appropriate observations and/over observation data over longer time periods is not mentioned); (iii) Possibilities/measures to reduce uncertainty, including past and future developments (e.g., numerical model simulations used in CAT models today); (iv) Perspectives: Challenges, further needs, and expected developments to address these needs (e e.g., role of crowd-sourced data).

2) Be more specific and give more details throughout the manuscript (see also minor points).

3) Loss/risk models are not appropriately described. In the hazard module, I miss the description of the (historic) event set, relevant parameters of the footprints/tracks (e.g., magnitude, width, length, orientation), and – based on this – stochastic modeling (indicated in Figure 1, but not mentioned in the text).

4) Refer to Solvency II and the need to assess probable maximum losses for 200-year return periods (PML200), as well as the need for a basic understanding of the models applied by the insurance companies.

5) Even though the insurance perspective is explicitly mentioned in the title, I miss a thorough discussion in the manuscript (see also point 1). Which perils are well represented by the models (and where), which are not? What are difficulties and challenges? What are new perspectives that might emerge in the future (e.g., role of machine learning / Big Data)? Some of these points are formulated as questions, but without providing answers or at least some hints (e.g., L109-111; L140-143).

6) In the context of global change, the manuscript only mentions climate change (very briefly) and increased population/wealth. However, global change has several other implications, such as the energy transition with an increased share of renewables with other vulnerabilities (e.g., solar panels are very susceptible to hail), increased reliance on critical infrastructure, or societal changes. All these issues have the potential to significantly change vulnerability and risk.

7) Section 5 is a summary rather than conclusions.

**Minor revision points:**

1. L4: the models assess **both** the risk of **experienced** events and **not yet experienced**

2. L10: "protect clients' property and activities"; it's rather risk transfer than protection
3. L16-19: maybe instead of formulations such as "unknowns unknowns", you may refer to their statistics? Further, is would be very helpful to learn more about how "unknowns unknowns" are considered by the insurance industry
4. L19-20: see comment 1 above
5. L30: explain "actuarial methods"; specify "extreme losses" (e.g., PML200, cf. major revisions point 4)
6. L34: "…whose impact was unexpected…": In what sense and why?
7. L42 briefly explain why each peril and region is usually modelled separately (you may refer here already to the global loss models suggested in the conclusions)
8. L45-47: I'm not sure what you mean by "format". If this refers to the data format only, then I would say that this problem is much easier solvable compared to the uncertainty inherent in each of the four model components (cf. major revision point 1).
9. L48-52: Is a storm like Hurricane Andrew accounted for in today's risk models, so has it turned from an "unknown unknown" to a "known known"?
10. L56-58: Mention that both monitoring of extremes as well as numerical modeling has substantially improved over the last decades leading to a better hazard estimation.
11. L60: "..notably the location at **high** granularity and the physical properties of building." Be more specific, give details; what granularity is required for what (exposure vs. hazard) depending on the different perils?
12. L66: "building damage" and "hazard magnitude" are two different topics; I wouldn't include both in one sentence.
13. L68-69: "It is less the case for other perils"; I cannot follow this statement, considering the devastation by, for example, tornadoes or hurricanes. "population is evacuated" is to general; evacuation is a measure in case of hurricanes, but usually not in case of windstorms, floods, or large hail.
14. L71 "Damage information…" Move this sentence to the beginning of this paragraph.
15. L80-81: this sentence is unclear (but becomes a bit clearer when reading the next sentence); I suggest to reformulate and to explicitly mention serial clustering at the beginning.
16. L84: you may also cite Vitolo et al. (2009, MZ, DOI 10.1127/0941-2948/2009/0393), the first paper on that topic
17. L88: Expand the discussion about uncertainty as this is highly relevant (cf. major point 1)
18. L96 and elsewhere: the expression "loss simulation **engine**" is strange. You mean a model? And why loss and not risk (if probability is considered in the hazard module)?
19. L98: explain "epistemic uncertainty"
20. 1st paragraph of Section 4m & Introduction: in the last sentence of the introduction, it was written that the paper focuses on the impact of natural hazards to **property exposure**. Section 4, however, describes supply chains and related interlinks. Of course, that topic is highly relevant for the general impact of natural hazards, but not for property exposure/loss.
21. L127: "shortages of cameras and smartphones". Even more important were shortages in HDs (hard discs) and chips reducing the overall computation power (cameras and smartphones at that time were mainly gadgets not generating real added value).
22. Check the references for consistency (e.g., some journals or manuscript titles are in bolt letters, other not)

**Edits:**

1. Check the appropriate use of \citep and \citet throughout the paper
2. L2: "undertaken" is not appropriate here
3. L41: "…here before cited…" needs rewording

4. L42: "peril x region" is unclear
5. L63: "all **being** critical…" loss**es**
6. L74: "…**to** collect…"
7. L83: " windstorms"; "**Serial** clustering" (note that there are different kinds of clustering, thus **serial** is important to include)
8. L85 process → probability
9. L86 exhibited in → shown by
10. L87: what do you mean by "dimensioning"?
11. L88: could → should
12. L115: "is intersected with hazard" → is interlinked with the hazard
13. L121/L124: "have become more interconnected" is mentioned twice here; further, mention the interrelation and dependencies of supply chains
14. L133: clients → companies; siloed → ?
15. L137: to which case study do you refer here?
16. L138: "exercise" is not an appropriate expression

---

## Author Comment (AC1)

**Answer to Referee Comment #1**

Thank you for your encouraging comments and the suggested modifications that will improve the quality of the paper and its readability. Please find below a point-by-point answer to the comments you raised. Please note that, as suggested by Referee #2 and #3 the structure of the paper will be rearranged. The updated structure is presented p4 of this document.

**Minor Corrections**

**Comment 1: Often loss models in the insurance industry are often referred to Catastrophe models. Was there a reason not to use this terminology in the paper?**

Thank you for raising that point. It is correct, and I should be more explicit on the reasons why I choose the term natural hazard modelling instead of catastrophe modelling.

Using the term catastrophe modelling implies that only events causing extreme damages are considered in the model. At the beginning, focus was made on high return-period loss events, driven by an extreme hazard intensity, an accumulation of exposure and/or vulnerable buildings. Catastrophe models were then used to assess the risk for a given portfolio. As more and more data has been included in the calibration of the models, they now capture much more than just extreme loss events. In the hazard module, all the spectrum of events (i.e. moderate/intense; frequent/rare events) is considered. In the vulnerability module, vulnerability curves cover all value of hazard intensity. In terms of usage, "catastrophe models" are now used to estimate budget and premium, mainly driven by smaller events, meaning that (re)insurers are interested in capturing small / frequent events in addition to large/rare ones.

I suggest evolving towards the use of the term natural hazard modelling as it widens the scope of events considered, compared to the term catastrophe modelling. In addition, as catastrophe models now include models for man-made perils (e.g. cyber risk, terrorism…), the term natural hazard modelling allows for more precision on the scope targeted by the model.

**Line 6. I'm not sure what this means can you rephrase "of the development of a wide community around natural hazards as well as of the occurrence of natural hazards."**

Agreed and modified

**Line 10. It would be useful to be specific and say "financial protection"?**

Insurance protection goes beyond the financial aspect of the risk transfer (i.e. the payment of a premium by the policy holder against the payment of future claims by the insurance company); it also includes prevention actions such as increasing risk awareness and proposing adapted protective solutions. I suggest to develop this point in the paper by giving two examples:

1) For the commercial business (corporates' policy holders): technical risk experts perform on-site visits to evaluate buildings' conditions and identify potential vulnerabilities to natural hazards. The objective is to assess how natural hazards could generate damage either to the building itself (e.g. storage warehouse, data center, shopping centers) or to its contents (e.g. machineries, production chain, products' stock…), and if such damages could cause business interruption (e.g. employees / clients cannot access the building for 10 days leading to a loss of turnover or profits). Following such assessment, prevention measures are then suggested or imposed to reduce the risk (e.g. elevate goods or machinery in the case of a flood event or reinforce some key components of the building to reduce the impact of ground shaking).
2) For the retail business (individual policy holders) : in this case, as protection actions cannot be tackled individually because of the mass of clients, they are taken globally. For example, after the Great Fire of London in 1666 that destroyed most of the buildings of the City -made of wood at that time-, insurance premium rates were lowered for building made of brick in order to encourage the use of bricks instead of wood and therefore reduce the fire risk in London.

**Line 11. Presumably you mean "insurance company" here?**

Agreed and modified

**Line 26. This sentence is very difficult to parse. I would suggest simplifying "keep refining what we know on one hand and, on the other hand, increasing insurers' preparedness to what we do not know" to "keep refining what we already know and to increase insurers preparedness for the unknown."**

Agreed and modified

**Line 37 Incorrect grammar. Perhaps rephrase "did not enable to seize the impact of growing exposure in particularly risk prone areas" to "were not able to assess the impact"**

Agreed and modified

**Line 41 "…work of characterization of the here before cited four components for various.." I'm not sure what this means can you rephrase.**

To give more clarity to this point, I have included more details on the loss modelling framework and each of the 4 components. This sentence is also rephrased.

**Line 42 Spelling mistake? "Each peril x region"**

Agreed and modified

**Line 43-47 The text here on the data formats used seems rather irrelevant for a brief history of loss modelling. There could be removed to make the paper more concise and improve readability, without affecting the main message of the manuscript.**

This is a key aspect of natural hazard modelling and one of the challenges faced today by the community. Data format is just the tip of the iceberg and refers to the way data is gathered and organized in each component of the loss modelling framework with the objectives of optimizing the run time (i.e. results are expected to be available after a few hours of run time) while dealing with IT constraints (i.e. memory limit, CPU/GPU…).

There is therefore a gap between the quality and the sophistication of modelling produced by research and the derivative data compiled to meet the requirements of the loss modelling framework. As an example, the severity of natural events is captured in the hazard component through the use of hazard footprints, defined as the maximum hazard value (e.g. windspeed, flood depth, peak ground acceleration…) at each grid cell of the considered area over the duration of the event. The information relative to the event duration and to the evolution of the hazard value over time are lost, while they are parameters that impact the assessment of buildings' damage.

As part of the restructuring of the paper (as presented in the supplement document), more details on why the transmission and the intersection of information from one component to the other is crucial.

**Line 48 "highlighted on one hand the non-modelled effects of the drivers of risk and on the other hand the insurance protection gap that was existing in Florida and the inefficiency of private and public mechanisms (McChristian, 2012)."**

**This sentence needs rephrasing – it's not clear what the "inefficiency of private and public mechanisms" is referring to.**

This will be rephrased.

**Line 60. I'm not sure what is meant here, are there missing words? "...notably the location at (longitude, latitude) granularity and the physical properties of buildings."**

More details will be given.

**Line 65 "observation data" should be "observational data"**

Agreed and modified

**Line 69. I'm not sure what is meant here, are there missing words? "…less structural damages on buildings and population is evacuated."**

Understood, it will be rephrased to explain better why there are less observational data of building damages following flood or windstorm events.

**Line 93 "This enables to identify sensitive components which may…" should be "This enables sensitive components to be identified, which may…"**

Agreed and modified

**Line 95. "Such an analysis requires first to run the production of the hazard catalogue several times…" should be "Such an analysis first requires the production of the hazard catalogue to be run several times…"**

Agreed and modified

**Line 116. "identified evolutions" perhaps should be "identified improvements"?**

Agreed and modified

**Line 127 Delete second repetition of "in the world" for readability.**

Deleted

**Line 128 Could you rephrase or expand on what is meant by "suppliers' default"?**

Agreed and rephrased. Suppliers' default, in the context of natural hazards, refers to the situation when a supplier is not able to provide its clients in the aftermath of a natural event.

**Line 131 "while making a research progress" should be "while making research progress"**

Agreed and modified

**Line 137 "insurers do not only need to" should be "insurers need to not only"**

Agreed and modified

**Line 137 The recent Fiedler et al. (2021) does a good job of outlining the challenges for climate change analytics and could be cited here.**

I completely agree and will refer to it.

**Line 147. "Over the years…" The sentence here is overlong and could be improved by splitting into two.**

Agreed and modified

**Line 153 Spelling mistake? "peril x region"**

Agreed and modified

**Suggested structure following RC2 and RC3 comments:**

(i)      Loss / risk model development from a historic perspective
-      Example of Hurricane Andrew
-      Detailed discussion on
   • the three components (hazard, exposure, vulnerability)
   • the loss simulation process (i.e. how the transmission and the intersection of information from one component to the other is performed)
-      Details related to the (re)insurance market and its evolution regarding natural hazard risk modelling

(ii)     Uncertainty in each of the components (current state) and its quantification / how we improve and measure what we know
-      Uncertainty driven by data quality and availability by component (exposure, hazard and vulnerability), some are inherent, some can be improved (e.g., uncertainty in hazard modelling due to a lack of appropriate observations and/over observation data over longer time periods is not mentioned). Include examples such as:
   • Improvement of exposure data to get precise information on buildings' coordinates and physical characteristics
   • The access to various type of hazard measurements, the availability of reanalysis datasets for atmospheric hazards
-      Uncertainty caused by modelling assumptions and approaches. Include examples such as:
   • Improvement of the modelling of serial clustering of European Windstorms
   • The impact of parameters setting in hydrologic tools (Kaczmarska et al. 2018)
-      Uncertainty caused by the implementation in the loss modelling framework

(iii)    Perspectives: Challenges, further needs, and expected developments to address these needs
-      Need for systematic analysis and quantification of uncertainties, component by component and on the overall loss simulation process
-      Identified challenges (e.g. how to model interrelated hazards and their impacts, how to model the impact of natural hazards on supply chain, the role of machine learning…)
-      Management of unknown unknowns in natural hazard modelling

---

## Author Comment (AC2)

**Answer to Referee Comment #2**

Thank you for your review and your interest in this paper. Answering your comments helped me to refine further the message I want to share in this paper and how to deliver them appropriately. Please find below a point-by-point answer to the comments you raised. Moreover, as suggested by Referee #3 and yourself, the structure of the paper will be rearranged. The suggested structure is presented p2 of this document.

1. **Either to change the title or, preferably, to widen the subject of the paper to the risk management of natural hazards. This latter would include besides the traditional mitigation strategies, also risk transfer and financing solutions.**

For more clarity the title will be changed, replacing "natural hazard risk management" by "natural hazard risk modelling". While it would be of interest to analyse current risk management solutions, I prefer to focus in this paper on the challenges ahead for the modelling of natural hazard that supports risk assessment and risk management.

2. **The paper has now the structure and the tone of a newspaper article. To be a scientific paper should: a) refer to data, b) be structured in a more rigorous and readable manner. For instance all the challenges mentioned could be structured referring to the different component, phases, of the risk modelling chain. I think the classification of knowns and unknowns is misleading given that in all the components, procedures, techniques and data used for risk modelling there's something already well consolidated and something not yet consolidated.**

I acknowledge that, by rearranging some parts, the message I want to convey will be clearer and the readability of the paper will be facilitated. Following your comment and the one from Referee #3, I suggest a new structure that is available in the supplement document.

Models' validation and quantification of uncertainties are key elements to reinforce and delineate the extent of what we know. However, regarding the terminology, I think that the terms "known" / "unknown" are more appropriate as the concept associated to these terms goes beyond the concept associated to "consolidated / not consolidated". Indeed, "known/unknowns" terminology encourages a mindset associated to becoming aware that a part of unknown will always be present, even though extensive validation and consolidation analyses are performed, and we need to deal with it. All the more in an ever-evolving environment (e.g. climate change, population migration, new data available, new techniques…) and given the complexity of the risk modelling, some results that have been consolidated in the past may become obsolete in the future.

3. **One of the most interesting feature of the paper is the perspective from an insurer. However there's very little presented from that perspective. Nonetheless the (re-)Insurance world has been completed reinvented in the last 20 years from many aspects: financially, regulatory, commercially and technically. The essence of the paper should be to tell to the scientific community the story of how the insurance sector has been changed by the possibility to quantify risk on each of those aspects and to write a list of open questions, a program for the next years to come for the scientific community on those aspects which can be of common interest with the insurance industry.**

The purpose of the paper is to present the current issues we face as insurers, opening up paths for researchers to define what they see as relevant and make sense within their research projects.

Regarding the lack of details on (re)insurance history, this will be tackled in the version of the paper, as part of the historical section as shown in the supplementary material.

**Suggested structure following RC2 and RC3 comments:**

(i)      Loss / risk model development from a historic perspective
- Example of Hurricane Andrew
- Detailed discussion on
  - the three components (hazard, exposure, vulnerability)
  - the loss simulation process (i.e. how the transmission and the intersection of information from one component to the other is performed)
- Details related to the (re)insurance market and its evolution regarding natural hazard risk modelling

(ii)     Uncertainty in each of the components (current state) and its quantification / how we improve and measure what we know
- Uncertainty driven by data quality and availability by component (exposure, hazard and vulnerability), some are inherent, some can be improved (e.g., uncertainty in hazard modelling due to a lack of appropriate observations and/over observation data over longer time periods is not mentioned). Include examples such as:
  - Improvement of exposure data to get precise information on buildings' coordinates and physical characteristics
  - The access to various type of hazard measurements, the availability of reanalysis datasets for atmospheric hazards
- Uncertainty caused by modelling assumptions and approaches. Include examples such as:
  - Improvement of the modelling of serial clustering of European Windstorms
  - The impact of parameters setting in hydrologic tools (Kaczmarska et al. 2018)
- Uncertainty caused by the implementation in the loss modelling framework

(iii)     Perspectives: Challenges, further needs, and expected developments to address these needs
- Need for systematic analysis and quantification of uncertainties, component by component and on the overall loss simulation process
- Identified challenges (e.g. how to model interrelated hazards and their impacts, how to model the impact of natural hazards on supply chain, the role of machine learning…)
- Management of unknown unknowns in natural hazard modelling

---

## Author Response (AR1)

**Author's response to referees' comments for NHESS-2022-6**

I thank the Editor Dr. David Peres for handling the manuscript and the three anonymous referees for their insightful comments.

In what follows, I show how the manuscript has been amended to take into account referees' comments. Compiling the initial comments from the three referees, this document includes:

- Referee comments in bold and black,
- My initial replies in black
- The modification made to the manuscript in red. Line numbers refer to the revised manuscript with changes not tracked.

**Answer to Referee Comment #1**

Thank you for your encouraging comments and the suggested modifications that will improve the quality of the paper and its readability. Please find below a point-by-point answer to the comments you raised. Please note that, as suggested by Referee #2 and #3 the structure of the paper will be rearranged. The suggested structure is presented in the supplement document.

**Minor Corrections**

**Comment 1: Often loss models in the insurance industry are often referred to Catastrophe models. Was there a reason not to use this terminology in the paper?**

Thank you for raising that point. It is correct, and I should be more explicit on the reasons why I choose the term natural hazard modelling instead of catastrophe modelling.

Using the term catastrophe modelling implies that only events causing extreme damages are considered in the model. At the beginning, focus was made on high return-period loss events, driven by an extreme hazard intensity, an accumulation of exposure and/or vulnerable buildings. Catastrophe models were then used to assess the risk for a given portfolio. As more and more data has been included in the calibration of the models, they now capture much more than just extreme loss events. In the hazard module, all the spectrum of events (i.e. moderate/intense; frequent/rare events) is considered. In the vulnerability module, vulnerability curves cover all value of hazard intensity. In terms of usage, "catastrophe models" are now used to estimate budget and premium, mainly driven by smaller events, meaning that (re)insurers are interested in capturing small / frequent events in addition to large/rare ones.

I suggest evolving towards the use of the term natural hazard modelling as it widens the scope of events considered, compared to the term catastrophe modelling. In addition, as catastrophe models now include models for man-made perils (e.g. cyber risk, terrorism…), the term natural hazard modelling allows for more precision on the scope targeted by the model.

Lines 98 – 105 : In the past few years, both the scope and use of catastrophe models have evolved. Indeed, to estimate the insurance premium of an average risk, insurers are now as interested in capturing small frequent events as large rare ones. As more and more data have been included in the calibration of the catastrophe models, they do capture much more than just extreme loss events. In the hazard module, the full spectrum of events (i.e. moderate/intense; frequent/rare events) is considered. In the vulnerability module, vulnerability curves cover the entire range of hazard intensity. As for the modelling scope, catastrophe models even exist for man-made perils such as cyber and terrorism. To reflect these evolutions, we will use the term 'natural hazard models / modelling' as it allows for greater precision on the model's targeted scope and reaffirms the use of these models for other purposes than the analysis of extreme events.

**Line 6. I'm not sure what this means can you rephrase "of the development of a wide community around natural hazards as well as of the occurrence of natural hazards."**

Agreed and modified

Lines 155 – 157: The (re)insurance market's current body of knowledge on natural 5 hazards loss modelling results from over 30 years' research involving private companies like (re)insurers, brokers, and modelling firms and academic researchers in atmospheric sciences, geosciences, civil engineering studies, and data sciences, to name but a few disciplines.

**Line 10. It would be useful to be specific and say "financial protection"?**

Insurance protection goes beyond the financial aspect of the risk transfer (i.e. the payment of a premium by the policy holder against the payment of future claims by the insurance company); it also includes prevention actions such as increasing risk awareness and proposing adapted protective solutions. I suggest to develop this point in the paper by giving two examples:

1) For the commercial business (corporates' policy holders): technical risk experts perform on-site visits to evaluate buildings' conditions and identify potential vulnerabilities to natural hazards. The objective is to assess how natural hazards could generate damage either to the building itself (e.g. storage warehouse, data center, shopping centers) or to its contents (e.g. machineries, production chain, products' stock…), and if such damages could cause business interruption (e.g. employees / clients cannot access the building for 10 days leading to a loss of turnover or profits). Following such assessment, prevention measures are then suggested or imposed to reduce the risk (e.g. elevate goods or machinery in the case of a flood event or reinforce some key components of the building to reduce the impact of ground shaking).

2) For the retail business (individual policy holders) : in this case, as protection actions cannot be tackled individually because of the mass of clients, they are taken globally. For example, after the Great Fire of London in 1666 that destroyed most of the buildings of the City -made of wood at that time-, insurance premium rates were lowered for building made of brick in order to encourage the use of bricks instead of wood and therefore reduce the fire risk in London.

Lines 14 – 26 :

Insurance protection goes beyond risk transfer (i.e., the payment of a premium against future claims); it also encompasses prevention actions such as reinforcing customers' risk awareness and proposing adapted protective solutions.

For example, in commercial business, technical risk experts perform on-site visits to evaluate the state of buildings and identify potential vulnerabilities to natural hazards. The objective is to assess how natural hazards could generate damage either to the buildings themselves (e.g. storage warehouses, data centers, shopping centers) or to their contents (e.g. machinery, production chains, stock), and if such damage could cause business interruption (e.g. employees / clients being unable to access the building for N days resulting in a loss of turnover or profits). Prevention measures like elevating goods or machinery in the event of flooding are then suggested or imposed post-assessment to reduce the risk and adjust the premium.

In the retail business with its mass of clients, protection actions have to be taken globally instead of individually. For example, after the Great Fire of London in 1666 that destroyed most of the city's buildings, made of wood at that time, insurance premium rates were lowered for buildings made of brick to encourage brick constructions instead of wood thus reducing the fire risk in London.

**Line 11. Presumably you mean "insurance company" here?**

Agreed and modified

Lines 27 – 28: To achieve their mission of protection, it is essential for insurers to identify and quantify the risks associated with the underwritten policies

**Line 26. This sentence is very difficult to parse. I would suggest simplifying "keep refining what we know on one hand and, on the other hand, increasing insurers' preparedness to what we do**

**not know" to "keep refining what we already know and to increase insurers preparedness for the unknown."**

Agreed and modified

Lines 47 – 49: Prior to focusing on the scientific and technical advances made to keep refining what we know about the risk drivers, exposure, hazard, and vulnerability, and how to increase insurers' preparedness for the unknown, it is important to recall how the reassessment process engendered the natural hazard loss modeling framework from the outset.

**Line 37 Incorrect grammar. Perhaps rephrase "did not enable to seize the impact of growing exposure in particularly risk prone areas" to "were not able to assess the impact"**

Agreed and modified

Lines 56 – 60: The data were thus too coarse to capture the three risk drivers' individual impact on the losses: the exposure (e.g. what if the exposure is located in a more / less risky area), the hazard itself (e.g. what portion of the losses are generated by a storm surge versus wind in the case of a tropical cyclone), the vulnerability (e.g. how effective the flood defenses, building codes are). As a result, while the data and the resulting modelling failed to take into account individual effects when assessing extreme losses, it did reflect the state of what was known by insurers and public authorities at that time.

**Line 41 "…work of characterization of the here before cited four components for various.." I'm not sure what this means can you rephrase.**

To give more clarity to this point, I have included more details on the loss modelling framework and each of the 4 components. This sentence is also rephrased.

Lines 107 – 153: Subsection '2.2 Natural hazard modelling framework' has been added and include a description of the loss modelling framework.

**Line 42 Spelling mistake? "Each peril x region"**

Agreed and modified

Lines 287: "by hazard and region"

**Line 43-47 The text here on the data formats used seems rather irrelevant for a brief history of loss modelling. There could be removed to make the paper more concise and improve readability, without affecting the main message of the manuscript.**

This is a key aspect of natural hazard modelling and one of the challenges faced today by the community. Data format is just the tip of the iceberg and refers to the way data is gathered and organized in each component of the loss modelling framework with the objectives of optimizing the run time (i.e. results are expected to be available after a few hours of run time) while dealing with IT constraints (i.e. memory limit, CPU/GPU…).

There is therefore a gap between the quality and the sophistication of modelling produced by research and the derivative data compiled to meet the requirements of the loss modelling framework. As an example, the severity of natural events is captured in the hazard component through the use of hazard footprints, defined as the maximum hazard value (e.g. windspeed, flood depth, peak ground acceleration…) at each grid cell of the considered area over the duration of the event. The information relative to the event duration and to the evolution of the hazard value over time are lost, while they are parameters that impact the assessment of buildings' damage.

As part of the restructuring of the paper (as presented in the supplement document), more details on why the transmission and the intersection of information from one component to the other is crucial.

Lines 107 – 153: Subsection '2.2 Natural hazard modelling framework' has been added and include a description of the loss modelling framework.

**Line 48 "highlighted on one hand the non-modelled effects of the drivers of risk and on the other hand the insurance protection gap that was existing in Florida and the inefficiency of private and public mechanisms (McChristian, 2012)."**

**This sentence needs rephrasing – it's not clear what the "inefficiency of private and public mechanisms" is referring to.**

This will be rephrased.

The comment on the inefficiency of private and public mechanisms has been removed to focus on the underestimation of the loss associated to an event such as Andrew.

Lines 62 – 67: According to McChristian (2012), before Hurricane Andrew, the loss assessment for an event of that strength was $4 to $5bn. This is three times lower than Hurricane Andrew's actual loss at $15 bn. Insurers underestimated their exposure as well as their exposure's vulnerability to such an event. McChristian (2012) also indicates that though recent loss history was adjusted to reflect current macro-economic trends, it has failed to capture the increasing population over coastal areas. In the aftermath of Hurricane Andrew, a collective realization grew for the need to both separately characterize the three drivers of the risk - exposure, hazard and vulnerability - and model their interconnections.

**Line 60. I'm not sure what is meant here, are there missing words? "...notably the location at (longitude, latitude) granularity and the physical properties of buildings."**

More details will be given.

Lines 168 – 174: Over the past 5 to 10 years, insurers have significantly improved the collection process of information characterizing their exposure, namely the coordinates of the location of the buildings as well as the buildings' physical properties. As mentioned previously, exposure data in the loss modelling process is used (i) to estimate the hazard's severity at the location of the building and (ii) to select the suitable damage curve. The more precise the exposure data, the more accurate the loss evaluation will be. However, as some elements are particularly difficult to get at the time of underwriting individual insurance, the systematic extraction and completion of the data remains a challenge and any missing information needs to be completed once the policy is underwritten either in the exposure database or at a later stage in the modelling.

**Line 65 "observation data" should be "observational data"**

Agreed and modified

Lines 159: granular observational data

**Line 69. I'm not sure what is meant here, are there missing words? "…less structural damages on buildings and population is evacuated."**

Understood, it will be rephrased to explain better why there are less observational data of building damages following flood or windstorm events.

Comment not kept in the final version of the paper.

**Line 93 "This enables to identify sensitive components which may…" should be "This enables sensitive components to be identified, which may…"**

Agreed and modified

Lines 197 – 205:

Beyond the perpetual enhancement necessary to complete and refine the view of the risk and to adapt to an ever-evolving environment, uncertainties persist in being only partially quantified due to (i) IT constraints and (ii) the information loss perpetuated by simplifying assumptions to derive data compiled in the loss modelling framework. Resolving these two sources of uncertainties would enable insurers to heighten their understanding of the risk and make sounder business decisions.

Uncertainties in the hazard component come from the input data and the modelling parameters used to generate the stochastic event catalogue. For example, Kaczmarska et al. (2018) quantify how in changing flooding parameters the loss estimates are impacted. Winter et al. (2018) go a step further notably in identifying and quantifying uncertainties present in the production of the hazard events catalogue.

**Line 95. "Such an analysis requires first to run the production of the hazard catalogue several times…" should be "Such an analysis first requires the production of the hazard catalogue to be run several times…"**

Agreed and modified

Lines 205 – 207:

Such an analysis first requires running the production of the hazard catalogue several times to test different sets of parameters and secondly running the loss simulation engine multiple times. Including the quantification of uncertainties is costly both in terms of computer power and runtime but should be systematized as a modelling best practice.

**Line 116. "identified evolutions" perhaps should be "identified improvements"?**

Agreed and modified

Lines 245 – 248: The shortcomings of the current loss modelling framework herein described point to the need for an in-depth review of the framework to improve and increase insurers' understanding of natural hazard risk particularly in an ever more connected environment that is described in the next section.

**Line 127 Delete second repetition of "in the world" for readability.**

Deleted

**Line 128 Could you rephrase or expand on what is meant by "suppliers' default"?**

Agreed and rephrased. Suppliers' default, in the context of natural hazards, refers to the situation when a supplier is not able to provide its clients in the aftermath of a natural event.

Lines 280 – 282: From an insurer's perspective, suppliers' defaulting in their deliveries due to the occurrence of a natural hazard is not insurable, as it is not quantifiable with the current modelling that fails to capture this connection between suppliers and their client producers.

**Line 131 "while making a research progress" should be "while making research progress"**

Agreed and modified

Lines 283 – 285: This could provide a source of opportunities for insurers to deliver new services to customers while continuing to contribute to advancing research in visualizing and measuring the levels of complexity (volume, direction and intensity of interconnections).

**Line 137 "insurers do not only need to" should be "insurers need to not only"**

Agreed and modified

Lines 294 – 295: To perform this exercise, insurers need to not only project the plausible future scenarios of hazard events (information provided by climate model simulations) but also to project the evolution of exposure and vulnerability.

**Line 137 The recent Fiedler et al. (2021) does a good job of outlining the challenges for climate change analytics and could be cited here.**

I completely agree and will refer to it.

Lines 302 – 304: As Fiedler et al. (2021) highlight, uncertainty around future exposure, hazard or vulnerability projections, is significant and compounds the uncertainty already present in the loss modelling framework.

**Line 147. "Over the years…" The sentence here is overlong and could be improved by splitting into two.**

Agreed and modified

Lines 306 – 309: To date, models have evolved through the incorporation of new information, without ever undergoing an in-depth transformation. Modifications have stemmed from the observation of the growing number of interconnections – and mutual impacts – at multiple levels: between insured customers and their suppliers and interactions and cross impacts between the disasters-causing natural phenomena.

**Line 153 Spelling mistake? "peril x region"**

Agreed and modified

Lines 287: "by hazard and region"

**Answer to Referee Comment #2**

Thank you for your review and your interest in this paper. Answering your comments helped me to refine further the message I want to share in this paper and how to deliver them appropriately. Please find below a point-by-point answer to the comments you raised. Moreover, as suggested by Referee #3 and yourself, the structure of the paper will be rearranged. The suggested structure is presented in the supplement document.

1. **Either to change the title or, preferably, to widen the subject of the paper to the risk management of natural hazards. This latter would include besides the traditional mitigation strategies, also risk transfer and financing solutions.**

For more clarity the title will be changed by replacing "natural hazard risk management" by "natural hazard risk modelling". While it would be of interest to analyse current risk management solutions, I prefer to focus in this paper on the challenges ahead for the modelling of natural hazard that supports risk assessment and risk management.

Updated title:

Invited perspectives: An insurer's perspective on the knowns and unknowns to face in natural hazard risk modelling

2. **The paper has now the structure and the tone of a newspaper article. To be a scientific paper should: a) refer to data, b) be structured in a more rigorous and readable manner. For instance all the challenges mentioned could be structured referring to the different component, phases, of the risk modelling chain. I think the classification of knowns and unknowns is misleading given that in all the components, procedures, techniques and data used for risk modelling there's something already well consolidated and something not yet consolidated.**

I acknowledge that, by rearranging some parts, the message I want to convey will be clearer and the readability of the paper will be facilitated. Following your comment and the one from Referee #3, I suggest a new structure that is available in the supplement document.

Models' validation and quantification of uncertainties are key elements to reinforce and delineate the extent of what we know. However, regarding the terminology, I think that the terms "known" / "unknown" are more appropriate as the concept associated to these terms goes beyond the concept associated to "consolidated / not consolidated". Indeed, "known/unknowns" terminology encourages a mindset associated to becoming aware that a part of unknown will always be present, even though extensive validation and consolidation analyses are performed, and we need to deal with it. All the more in an ever-evolving environment (e.g. climate change, population migration, new data available, new techniques…) and given the complexity of the risk modelling, some results that have been consolidated in the past may become obsolete in the future.

**Revised structure following RC2 and RC3 comments:**

1 Introduction

2 Natural hazard modelling: a brief overview

      2.1 The co-influence of (re)insurance market and natural hazard modelling

      2.2 Natural hazard modelling framework

3 Current challenges in modelling natural hazards

      3.1 Exposure component

      3.2 Hazard component

3. **One of the most interesting feature of the paper is the perspective from an insurer. However there's very little presented from that perspective. Nonetheless the (re-)Insurance world has been completed reinvented in the last 20 years from many aspects: financially, regulatory, commercially and technically. The essence of the paper should be to tell to the scientific community the story of how the insurance sector has been changed by the possibility to quantify risk on each of those aspects and to write a list of open questions, a program for the next years to come for the scientific community on those aspects which can be of common interest with the insurance industry.**

The purpose of the paper is to present the current issues we face as insurers, opening up paths for researchers to define what they see as relevant and make sense within their research projects.

In section 4. 'Future challenges and further needs', three areas of research that would advance natural risk assessment and support insurers' in their ambition to more accurately project and plan out their business activities related to natural hazards.

- Introducing a fifth component to quantify the uncertainty
- Supply chain modelling
- Forward-looking scenario: modelling the future of natural hazard risks

Regarding the lack of details on (re)insurance history, this will be tackled in the version of the paper, as part of the historical section as shown in the supplementary material.

(Re)insurance history is presented in subsection 2.1 'The co-influence of (re)insurance market and natural hazard modelling', along with the history of natural hazard modelling

**Answer to Referee Comment #3**

Thank you for your review and your suggestions that will improve the readability of the paper. Please find below a point-by-point answer to the comments you raised. Moreover, as suggested by Referee #2 and yourself, the structure of the paper will be rearranged. The suggested structure is presented in the supplement document.

**Major revision points:**

**1) I suggest revising the structure of the manuscript. The current version sometimes jumps from one topic to another and sometimes back again (e.g., paragraph 65-87); some statements refer only to individual hazards (e.g., flooding), while the next sentence is a general statement.**

**My suggestion is the following structure: (i) Loss / risk model development from a historic perspective, including a detailed discussion of the three components (hazard, exposure, vulnerability); (ii) Uncertainty inherent in each of the components (e.g., uncertainty in hazard modeling due to a lack of appropriate observations and/over observation data over longer time periods is not mentioned); (iii) Possibilities/measures to reduce uncertainty, including past and future developments (e.g., numerical model simulations used in CAT models today); (iv) Perspectives: Challenges, further needs, and expected developments to address these needs (e e.g., role of crowd-sourced data).**

Thank you for suggesting this structure for the paper. This will indeed bring more clarity and strengthen the link between the different topics.

I would propose the following modifications:

(i)      Loss / risk model development from a historic perspective, including a detailed discussion of the three components (hazard, exposure, vulnerability);
          Include also details on how the transmission and the intersection of information from one component to the other is performed and is crucial in how uncertainty is propagated along the modelling chain
          Include details related to (re)insurance market

(ii)      Uncertainty inherent in each of the components (e.g., uncertainty in hazard modeling due to a lack of appropriate observations and/over observation data over longer time periods is not mentioned);
          I would suggest addressing uncertainty at large, not focusing only on uncertainties inherent to the modelling. Uncertainties could be discussed along 3 axes:
             Uncertainty driven by data quality and availability by component (exposure, hazard and vulnerability), some are inherent, some can be improved
             Uncertainty caused by modelling assumptions and approaches
             Uncertainty driven by the implementation in the loss modelling framework

(iii)      Possibilities/measures to reduce uncertainty, including past and future developments (e.g., numerical model simulations used in CAT models today);
          I would suggest splitting this section into the (ii) and the (iv) sections as the past developments would be examples of observed and resolved uncertainties; and the future developments will be addressed as expected developments to meet identified limitations in current modelling.

(iv)      Perspectives: Challenges, further needs, and expected developments to address these needs (e e.g., role of crowd-sourced data).
          Ok

**Revised structure following RC2 and RC3 comments:**

1 Introduction

**2) Be more specific and give more details throughout the manuscript (see also minor points).**

Agreed.

**3) Loss/risk models are not appropriately described. In the hazard module, I miss the description of the (historic) event set, relevant parameters of the footprints/tracks (e.g., magnitude, width, length, orientation), and – based on this – stochastic modeling (indicated in Figure 1, but not mentioned in the text).**

A description on the three components, as well as how the interconnexion is performed, will be added.

Lines 107 – 153: Subsection '2.2 Natural hazard modelling framework' has been added and include a description of the loss modelling framework.

**4) Refer to Solvency II and the need to assess probable maximum losses for 200-year return periods (PML200), as well as the need for a basic understanding of the models applied by the insurance companies.**

Agreed, I will include this in the introduction, or the first section defined in comment 1. While Solvency 2 accelerated the use of natural hazard models within the insurance industry to evaluate the risk (i.e. focus on extreme losses at portfolio level), the challenge for the insurance industry is now to connect these models to their pricing tools, for which average loss at a building level is used.

(Re)insurance history and regulatory environment are presented in subsection 2.1 'The co-influence of (re)insurance market and natural hazard modelling', along with the history of natural hazard modelling

**5) Even though the insurance perspective is explicitly mentioned in the title, I miss a thorough discussion in the manuscript (see also point 1). Which perils are well represented by the models (and where), which are not? What are difficulties and challenges? What are new perspectives that might emerge in the future (e.g., role of machine learning / Big Data)? Some of these points are formulated as questions, but without providing answers or at least some hints (e.g., L109-111; L140-143).**

Could you please precise the question "Which perils are well represented by the models (and where), which are not?"?

Do you mean how well are the perils represented in terms of hazard and frequency in the hazard component? or do you mean how well the perils are represented in terms of losses?

Lines 196 – 201: A review of hazard modelling approaches by peril can be found in Ward et al. (2020). Beyond the perpetual enhancement necessary to complete and refine the view of the risk and to adapt to an ever-evolving environment, uncertainties persist in being only partially quantified due to (i) IT constraints and (ii) the information loss perpetuated by simplifying assumptions to derive data compiled in the loss modelling framework. Resolving these two sources of uncertainties would enable insurers to heighten their understanding of the risk and make sounder business decisions.

**6) In the context of global change, the manuscript only mentions climate change (very briefly) and increased population/wealth. However, global change has several other implications, such as the energy transition with an increased share of renewables with other vulnerabilities (e.g., solar panels are very susceptible to hail), increased reliance on critical infrastructure, or societal changes. All these issues have the potential to significantly change vulnerability and risk.**

I totally agree and will adapt the paper accordingly. I will also add details on the importance of prevention / protection measures in the reduction of the impact of natural hazards.

**7) Section 5 is a summary rather than conclusions.**

In the light of the suggested new structure, this section is no longer needed.

**Minor revision points:**

**1. L4: the models assess both the risk of experienced events and not yet experienced**

Is it an affirmation or a suggestion to rephrase? What do you mean by models assess the risk of experienced events?

**2. L10: "protect clients' property and activities"; it's rather risk transfer than protection**

Insurance protection goes beyond the financial aspect of the risk transfer (i.e. the payment of a premium by the policy holder against the payment of future claims by the insurance company); it also includes prevention actions such as increasing risk awareness and proposing adapted protective solutions. I suggest to develop this point in the paper by giving two examples:

1) For the commercial business (corporates' policy holders): technical risk experts perform on-site visits to evaluate buildings' conditions and identify potential vulnerabilities to natural hazards. The objective is to assess how natural hazards could generate damage either to the building itself (e.g. storage warehouse, data center, shopping centers) or to its contents (e.g. machineries, production chain, products' stock…), and if such damages could cause business interruption (e.g. employees / clients cannot access the building for 10 days leading to a loss of turnover or profits). Following such assessment, prevention measures are then suggested or imposed to reduce the risk (e.g. elevate goods or machinery in the case of a flood event or reinforce some key components of the building to reduce the impact of ground shaking).
2) For the retail business (individual policy holders) : in this case, as protection actions cannot be tackled individually because of the mass of clients, they are taken globally. For example, after the Great Fire of London in 1666 that destroyed most of the buildings of the City -made of wood at that time-, insurance premium rates were lowered for building made of brick in order to encourage the use of bricks instead of wood and therefore reduce the fire risk in London.

Lines 14 – 26:

Insurance protection goes beyond risk transfer (i.e., the payment of a premium against future claims); it also encompasses prevention actions such as reinforcing customers' risk awareness and proposing adapted protective solutions.

For example, in commercial business, technical risk experts perform on-site visits to evaluate the state of buildings and identify potential vulnerabilities to natural hazards. The objective is to assess how

natural hazards could generate damage either to the buildings themselves (e.g. storage warehouses, data centers, shopping centers) or to their contents (e.g. machinery, production chains, stock), and if such damage could cause business interruption (e.g. employees / clients being unable to access the building for N days resulting in a loss of turnover or profits). Prevention measures like elevating goods or machinery in the event of flooding are then suggested or imposed post-assessment to reduce the risk and adjust the premium.

In the retail business with its mass of clients, protection actions have to be taken globally instead of individually. For example, after the Great Fire of London in 1666 that destroyed most of the city's buildings, made of wood at that time, insurance premium rates were lowered for buildings made of brick to encourage brick constructions instead of wood thus reducing the fire risk in London.

**3. L16-19: maybe instead of formulations such as "unknowns unknowns", you may refer to their statistics? Further, is would be very helpful to learn more about how "unknowns unknowns" are considered by the insurance industry**

When statistical metrics can be assessed to measure uncertainty, it means that it is possible to delineate the extent of what we know, but not necessarily to capture what we still don't know. Implementing statistical measure of uncertainty would be already a significant step for natural hazard modelling and (re)insurers.

Unknown unknowns are usually dealt through the definition and quantification of scenarios combining several simultaneous and adverse situations. The objective is then to test the robustness and limitation of the risk management solutions put in place.

**4. L19-20: see comment 1 above**

Answered in comment 1.

**5. L30: explain "actuarial methods"; specify "extreme losses" (e.g., PML200, cf. major revisions point 4)**

Agreed and included in the revision of the paper. In the context of natural hazard, actuarial methods refer to statistical functions used with the objective to estimate the value at risk of a given portfolio. Losses are assessed as extreme when their probability of occurrence is higher than the quantile 99.5.

Lines 85 – 97: The implementation of regulation has prompted insurers to use catastrophe models, mainly licensed by third-party vendors, as tools to assess the risk, define the risk appetite, and set the solvency capital requirement. For example, the Solvency 2 regulation implemented in Europe in 2016 requires (re)insurers with Europe-based headquarters to annually assess their loss for a 200-year return period shock. (Re)Insurers conduct this assessment for all the risks they are exposed to. They then aggregate these estimated losses to determine the total potential loss and the economic capital they have to bear in their owned funds. To achieve this assessment, (re)insurers have two options: either to use the so-called Standard Formula, calibrated on market exposure, or to develop an internal view of their risk that requires regulator approval. Most (re)insurers choosing to develop their own view of natural hazard risk use one or several models licensed to third-party vendors; others develop their own suite of models. When (re)insurers opt for using third-party models, model evaluation becomes a necessary activity for assessing the model's strengths and limitations leads to gaining in understanding and in taking ownership of the model. Model adjustments may be defined and applied to the models' loss estimation to address identified limitations (e.g. a non-modeled peril). (Re)insurance companies also invest in the development of in-house models either on scopes where no third-party vendors model is available or to gain in flexibility and transparency.

**6. L34: "…whose impact was unexpected…": In what sense and why?**

According to McChristian (2012), before the occurrence of Hurricane Andrew, the loss assessment for an event of this strength was $4 to $5 bn, which is 3 times lower than the actual loss of Hurricane

Andrew at $15 bn. Insurers underestimated their exposure at risk as well as its vulnerability to such an event. The author also indicates that recent loss history was adjusted to reflect current macro-economic trends and did not capture the increasing population over coastal areas. Historical loss data were too coarse to capture the separated impact on losses driven by storm surge versus wind, nor the impact driven by growing exposure or a change in building codes. Consequently, statistical models used to extrapolate the historical losses record to assess more extreme losses could not take these effects into account either.

Lines 62 to 67: According to McChristian (2012), before Hurricane Andrew, the loss assessment for an event of that strength was $4 to $5bn. This is three times lower than Hurricane Andrew's actual loss at $15 bn. Insurers underestimated their exposure as well as their exposure's vulnerability to such an event. McChristian (2012) also indicates that though recent loss history was adjusted to reflect current macro-economic trends, it has failed to capture the increasing population over coastal areas. In the aftermath of Hurricane Andrew, a collective realization grew for the need to both separately characterize the three drivers of the risk - exposure, hazard and vulnerability - and model their interconnections.

**7. L42 briefly explain why each peril and region is usually modelled separately (you may refer here already to the global loss models suggested in the conclusions)**

To be included.

The development of natural hazards models by region and peril has been mainly opportunistic, driven by (re)insurance market exposure and associated risk.

**8. L45-47: I'm not sure what you mean by "format". If this refers to the data format only, then I would say that this problem is much easier solvable compared to the uncertainty inherent in each of the four model components (cf. major revision point 1).**

The implementation of research within components of the loss modelling framework is a key aspect of natural hazard modelling and one of the challenges faced today by the community. Data format is just the tip of the iceberg and refers to the way data is gathered and organized in each component of the loss modelling framework with the objectives of optimizing the run time (i.e. results are expected to be available after a few hours of run time) while dealing with IT constraints (i.e. memory limit, CPU/GPU…).

There is therefore a gap between the quality and the sophistication of modelling produced by research and the derivative data compiled to meet the requirements of the loss modelling framework. As an example, the severity of natural events is captured in the hazard component through the use of hazard footprints, defined as the maximum hazard value (e.g. windspeed, flood depth, peak ground acceleration…) at each grid cell of the considered area over the duration of the event. The information relative to the event duration and to the evolution of the hazard value over time are lost, while they are parameters that impact the assessment of buildings' damage.

As part of the restructuring of the paper (as presented in the supplement document), more details on why the transmission and the intersection of information from one component to the other is crucial.

Lines 107 – 153: Subsection '2.2 Natural hazard modelling framework' has been added and include a description of the loss modelling framework.

**9. L48-52: Is a storm like Hurricane Andrew accounted for in today's risk models, so has it turned from an "unknown unknown" to a "known known"?**

The modelling approach by component developed in the aftermath of Hurricane Andrew remedied to the limitation of models at the time that did not consider the non-linear impact of growing exposure in exposed areas. I would say that, nowadays, models can reproduce quite precisely the impacts generated by Hurricane Andrew at that time. However, since the occurrence of Andrew, there have been evolutions of the local environment that are not captured by models today. For example:

- soil erosion or the sinking of coastal cities such as Miami may increase the impact of hurricanes
- the reinforcement of mangroves along the coastline may decrease the impact of hurricanes

Hurricane Andrew impacts in 1992 are known knowns as we have data and models that can reproduce it. This does not mean that Hurricane Andrew-like event in 2022 are known knowns.

**10. L56-58: Mention that both monitoring of extremes as well as numerical modeling has substantially improved over the last decades leading to a better hazard estimation.**

Agreed.

Lines 193 – 195: An ever-growing amount of data on the hazard component has been made accessible, refined, and maintained. A multitude of types of data, from observations to model simulations or a mixture of both, substantially support the development of hazard catalogues and their validation.

**11. L60: "..notably the location at high granularity and the physical properties of building." Be more specific, give details; what granularity is required for what (exposure vs. hazard) depending on the different perils?**

This point will be integrated in the changes suggested in major comment 1 and the description of the 4 components.

Lines 107 – 153: Subsection '2.2 Natural hazard modelling framework' has been added and include a description of the loss modelling framework.

**12. L66: "building damage" and "hazard magnitude" are two different topics; I wouldn't include both in one sentence.**

Hazard magnitude might be a shortcut. What I mean is that to improve the loss modelling, in particular the vulnerability component, we need to collect the information on:

(i) how severe was the event at the location of the building (i.e. values of the relevant hazards' variables leading to building's damage)
(ii) how damaged is the building (including also all relevant information on the building itself)
(iii) what were the associated repairing costs

Lines 229 – 234: While there has been a substantial increase in the availability of observational data over the past two decades (Yu et al., 2018), further investments should be made to systematically collect: (i) the event's level of hazard severity at the building's location (i.e. values of the relevant hazards' variables leading to the building's damage), (ii) the building's level of damage and the prevention measures if any (concurrently recording all relevant information on the building itself) and (iii) the level of associated repair costs (including information on loss adjustments and economic metrics such as post event inflation).

**13. L68-69: "It is less the case for other perils"; I cannot follow this statement, considering the devastation by, for example, tornadoes or hurricanes. "population is evacuated" is too general; evacuation is a measure in case of hurricanes, but usually not in case of windstorms, floods, or large hail.**

Agreed, I will rephrase and clarify.

Removed in the revised version of the paper

**14. L71 "Damage information..." Move this sentence to the beginning of this paragraph.**

Agreed and modified.

Lines 225 – 227: Systematic data collection of damage information and its associated hazard magnitude is therefore vital to characterize the impact of natural hazards on buildings and to improve the calibrations not only of the buildings' destruction rate but also of the reconstruction costs in the vulnerability component

**15. L80-81: this sentence is unclear (but becomes a bit clearer when reading the next sentence); I suggest to reformulate and to explicitly mention serial clustering at the beginning.**

Agreed, it could be rephrased as follows:

As the occurrence of natural events brings new information and data, it is integrated into the loss modelling framework to improve the assessment of loss. For example, the modelling of serial clustering of European windstorms has greatly improved following the occurrence of Lothar and Martin in 1999.

Lines 71 – 74: The occurrence of natural disasters, in particular those with a strong impact for the (re)insurance market, continues to feed research insofar that the research is in turn integrated into the hazard and vulnerability components of the loss modelling framework every 2 to 5 years.

**16. L84: you may also cite Vitolo et al. (2009, MZ, DOI 10.1127/0941-2948/2009/0393), the first paper on that topic**

Agreed.

Lines 75 to 76: The serial clustering effect refers to the higher probability that two extreme windstorms occur in a short period of time, under particular atmospheric conditions (Vitolo et al., 2009; Pinto et al., 2013; Priestley et al., 2017).

**17. L88: Expand the discussion about uncertainty as this is highly relevant (cf. major point 1)**

This point will be integrated in the changes suggested in major comment 1.

Lines 259 – 273:

4.1 Introducing a fifth component to quantify the uncertainty

As stated in previous sections, the assessment of uncertainty all along the modelling chain constitutes the loss modelling framework's notable shortcoming and the one that requires further investigation. To a certain extent, uncertainty is inherent to modelling and is partly captured in the loss modelling framework today through (i) the primary uncertainty, that is the assumptions and the simulation of the hazard catalogue, and (ii) the secondary uncertainty, that is the damage and loss assessment.

Including the quantification of uncertainties in the loss modelling framework is costly both in terms of computer power and runtime. However, given the rapid evolution of IT, computer power and run time should not be an issue for long and the question will then be how to implement a comprehensive uncertainty quantification scheme. While Beven et al. (2018) suggest a framework to deal with epistemic uncertainty in natural hazard modelling, recent work like (Noacco et al., 2019; KC et al., 2020) has been carried out to address quantifying uncertainty with appropriate methods and tools. How about introducing a specific "uncertainty component" that would deal with the multiple datasets from the different components and propagate the quantification all along the loss modelling process?

Along with their systematic quantification, we are convinced that uncertainties' management and communication around it will evolve and that insurers will take ownership of this management and make it a tool to enhance the modelling (Thompson and Warmink, 2016; Doyle et al., 2019).

**18. L96 and elsewhere: the expression "loss simulation engine" is strange. You mean a model? And why loss and not risk (if probability is considered in the hazard module)?**

Agreed. I would rather use the expression "loss simulation process" to designate the process that performs the loss assessment. For each building of the insurance portfolio and for each event of the hazard component:

1. building's location available in the exposure component is intersected with the event footprint to obtain the hazard intensity value at the location of the building.

2. based on the hazard intensity value and the physical properties of the building, the corresponding damage ratio is derived using the vulnerability curve associated to the characteristics of the building.

3. the damage ratio is applied to the sum insured of the building, as given in the financial module, which results into a loss amount. Financial conditions are applied to the loss amount to get the ultimate loss borne by the insurance company.

The loss simulation process produces Exceedance Probability curves characterizing the risk, i.e. the probability to exceed an amount of loss.

Use of the term 'loss modelling process' throughout the paper

**19. L98: explain "epistemic uncertainty"**

Epistemic uncertainty is the uncertainty due to lack of information or knowledge of the hazard, in particular because historical observations are not sufficient to capture the complexity of the hazard.

**20. 1 st paragraph of Section 4m & Introduction: in the last sentence of the introduction, it was written that the paper focuses on the impact of natural hazards to property exposure. Section 4, however, describes supply chains and related interlinks. Of course, that topic is highly relevant for the general impact of natural hazards, but not for property exposure/loss.**

Indeed, while Business Interruption (BI) following natural events is included in insurance contracts, Non-Direct Business Interruption (NDBI) is usually excluded. The point here is that by investigating this type of exposure and the risk associated to it, it might become possible to include it.

**21. L127: "shortages of cameras and smartphones". Even more important were shortages in HDs (hard discs) and chips reducing the overall computation power (cameras and smartphones at that time were mainly gadgets not generating real added value).**

Thank you for that comment, I will mention that as well.

Lines 276 – 279: This dependency became apparent with the 2011 floods in Thailand when Thailand's brutal interruption of microprocessor production led to a halt in global production, a global shortage of microprocessors, and consequently, a loss in benefits for companies producing chips, hard disc drives and other electronic devices (Chopra and Sodhi, 2014; Haraguchi and Lall, 2015).

**22. Check the references for consistency (e.g., some journals or manuscript titles are in bolt letters, other not)**

Agreed and modified.

**Edits:**

**1. Check the appropriate use of \citep and \citet throughout the paper**

Agreed and corrected.

**2. L2: "undertaken" is not appropriate here**

Corrected

Line 2: all the underwritten risks

**3. L41: "…here before cited…" needs rewording**

It has been reworded

Removed in the revised version of the paper

**4. L42: "peril x region" is unclear**

Replaced by "each scope, defined by one peril and one region,"

Removed in the revised version of the paper

**5. L63: "all being critical…" losses**

Corrected

Lines 176 – 177: With an address, it is possible to get the geolocation, the structure of the building, number of floors and even the roof type, all critical drivers of damage for different perils

**6. L74: "…to collect…"**

Corrected

Lines 227 – 228: New technologies such as drones and satellites provide alternative ways to access impacted areas to collect detailed and granular measurements

**7. L83: "winter windstorms"; "Serial clustering" (note that there are different kinds of clustering, thus serial is important to include)**

Corrected

Lines 73 – 78: This is how, the successive 1999 occurrence of the two extreme European winter windstorms Lothar and Martin triggered the introduction of the serial clustering effect in modelling the frequency of European winter windstorms (Mitchell-Wallace, 2017). The serial clustering effect refers to the higher probability that two extreme winter windstorms occur in a short period of time, under particular atmospheric conditions (Vitolo et al., 2009; Pinto et al., 2013; Priestley et al., 2017). Prior to these windstorms, the assumption used to calculate the occurrence probability of European winter windstorms followed the Poisson distribution and thus failed to allow for the modelling of successive events.

**8. L85 process > probability**

Corrected

Lines 76 – 78: Prior to these windstorms, the assumption used to calculate the occurrence probability of European windstorms followed the Poisson distribution and thus failed to allow for the modelling of successive events.

**9. L86 exhibited in > shown by**

Corrected

Lines 78 – 80: As shown by Priestley et al. (2018), the clustering effect has a significant impact on the estimation of yearly aggregated losses and therefore on the dimensioning of reinsurance covers.

**10. L87: what do you mean by "dimensioning"?**

Sizing

Lines 79: Sizing

**11. L88: could > should**

Corrected

Lines 260 – 261: As stated in previous sections, the assessment of uncertainty all along the modelling chain constitutes the loss modelling framework's notable shortcoming and the one that requires further investigation.

**12. L115: "is intersected with hazard" > is interlinked with the hazard**

Corrected

Removed in the revised version of the paper

**13. L121/L124: "have become more interconnected" is mentioned twice here; further, mention the interrelation and dependencies of supply chains**

Agreed.

Lines 275 – 276: With globalization, clients around the world have become increasingly interconnected and dependent on each other within so-called Global Value Chains (Gereffi et al., 2001; Baldwin and Lopez-Gonzalez, 2015; Phillips, 2018).

**14. L133: clients > companies; siloed > ?**

To be rephrased.

In the revised version, clients is not mentioned anymore and siloed is kept to

**15. L137: to which case study do you refer here?**

Rephrased "Another way to tackle unknown unknowns is to develop forward-looking views of the risk, as it is done in studies quantifying the impact of climate change on insurers' business."

Removed in the revised version of the paper

**16. L138: "exercise" is not an appropriate expression**

Replaced by "analysis"

Lines 294 and 302 :  Replaced by "analysis"

---

## Referee Report (RR1)

**Review of "Invited perspectives: An insurer's perspective on the knowns and unknowns to face in natural hazard risk modelling." By Madeline-Sophie Déroche**

The paper provides i) an overview of current practices in NH (natural hazard) risk modelling in the insurance sector and ii) outlines some of the grand challenges that will need to be solved in order to make advances in NH risk modelling. Most of the paper focuses on the overview of the current practices, and rather less on the future challenges, which I think reduces the originality and potential impact of the paper. Nonetheless, since the paper is written from the perspective of the insurance industry, I imagine that it will be of wide interest to the wider community of NH researchers who would like some insight into the challenges and opportunities face by the insurance industry. Subsequently, I would recommend the paper be published subject to the comments below being addressed.

The following comments are not "major" in the sense of needing a major revision. A revised version of the manuscript should address each of these points.

1. The abstract reads more like a short introduction than an abstract. I appreciate that there are word limits, but it would be good to reword the abstract to cut down on the introductory sentences and to at least mention the future challenges facing NH risk modelling that are described later in the paper.

2. Section 4.1: I fully agree that quantifying uncertainty is one of the grand challenges facing NH risk modelling in the insurance industry. However, a key component of assessing uncertainty is having transparent and openly accessible risk models that can be compared and evaluated. This isn't the norm in much of the insurance industry (important activities such as OASIS LMF being the exception). I think it would be very important to point this out in section 4.1.

3. There are a very large number of minor errors in the manuscript (e.g. repeated lines of text). I reviewed the Track Changes version of the revised manuscript as I could not find another version easily on the website, so perhaps there is another version with less errors. I would recommend that a revised manuscript receive a rigorous proof-read before being resubmitted.

**Minor Comments**

Title: I would suggest removing "to face" from the title for readability.

Line 14 "brokers and modelling firms" remove unnecessary "and" from the list

Line 21-24 This sentence is too long and difficult to read. It is also ambiguous, what are the two business sgements?

Line 26 A paragraph break is not needed here.

Line 40-41 "be they natural hazards, financial or cyber" is repeated twice, which makes the sentences here very difficult to read. Please simplify the sentence structure to avoid repetition.

Line 57 "dire" a better word here would be "important"

Line 113 "(e.g. non-modeled loss)" Do you have an example of what such a non-modeled loss would be?

Line 116 Replace with "As more data.." for readability

Line 116, 117 It's not clear what is meant here. You need more data to look at extreme events, so the meaning is not clear. Can you rephrase?

Line 156 "was" should be "is"

Line 171 Should it be two or three components? Only two reasons are listed in the rest of the sentence.

Line 179 Repeated lines

Line 185-190 I'm not sure what is meant in this paragraph. What are these gaps?

Line 214 Repeated lines

Line 216 What is the nature of the these sensitivity tests?

Line 227 Repeated lines

Line 250 "ultimate" final would be a better word

Line 254 "rupture" damage would be a better word.

Line 272 Repeated lines

Line 289 Repeated lines

Line 296 "societal matter" I'm not sure what is meant by this – can you clarify. It seems to more about modelling and data handling in insurance than a matter for wider society.

Line 300. "modelling of loss modelling frameworks" should be plural as you're speaking more generally

Line 315. I'm assuming your talking about ensemble modelling here? If not, can you be more specific what you are proposing for this "uncertainty component"

Section 4.3 It is important to explicitly use the words "climate change" in this section

---

## Author Response (AR2)

**Review of "Invited perspectives: An insurer's perspective on the knowns and unknowns to face in natural hazard risk modelling." By Madeline-Sophie Déroche**

The paper provides i) an overview of current practices in NH (natural hazard) risk modelling in the insurance sector and ii) outlines some of the grand challenges that will need to be solved in order to make advances in NH risk modelling. Most of the paper focuses on the overview of the current practices, and rather less on the future challenges, which I think reduces the originality and potential impact of the paper. Nonetheless, since the paper is written from the perspective of the insurance industry, I imagine that it will be of wide interest to the wider community of NH researchers who would like some insight into the challenges and opportunities face by the insurance industry. Subsequently, I would recommend the paper be published subject to the comments below being addressed.

The following comments are not "major" in the sense of needing a major revision. A revised version of the manuscript should address each of these points.

1. The abstract reads more like a short introduction than an abstract. I appreciate that there are word limits, but it would be good to reword the abstract to cut down on the introductory sentences and to at least mention the future challenges facing NH risk modelling that are described later in the paper.

Agreed, the abstract has been rewritten.

Abstract has been reworded accordingly

2. Section 4.1: I fully agree that quantifying uncertainty is one of the grand challenges facing NH risk modelling in the insurance industry. However, a key component of assessing uncertainty is having transparent and openly accessible risk models that can be compared and evaluated. This isn't the norm in much of the insurance industry (important activities such as OASIS LMF being the exception). I think it would be very important to point this out in section 4.1.

I have added a comment on this in the new version. While I agree on the necessity of having more transparent models and loss modelling tools, uncertainty quantification is not done at all, even by private modelling firms. I believe the natural hazard modelling community has to systematically quantify and communicate on uncertainty and sensitivity tests to assess the impact on the losses.

3. There are a very large number of minor errors in the manuscript (e.g. repeated lines of text). I reviewed the Track Changes version of the revised manuscript as I could not find another version easily on the website, so perhaps there is another version with less errors. I would recommend that a revised manuscript receive a rigorous proof-read before being resubmitted.

Minor Comments

Title: I would suggest removing "to face" from the title for readability.

Agreed and modified

Line 14 "brokers and modelling firms" remove unnecessary "and" from the list

Agreed and modified

Line 21-24 This sentence is too long and difficult to read. It is also ambiguous, what are the two business sgements?

Rephrased: Though insurers develop ever-increasing products to respond to clients' specific needs, P&C insurance in essence consists of two segments, the (i) retail business for home and car owners and (ii) commercial business for corporate clients.

Line 26 A paragraph break is not needed here.

Agreed and modified .

Line 40-41 "be they natural hazards, financial or cyber" is repeated twice, which makes the sentences here very difficult to read. Please simplify the sentence structure to avoid repetition.

The repetition does not appear in this version

Line 57 "dire" a better word here would be "important"

"Dire" replaced with "pressing"

Line 113 "(e.g. non-modeled loss)" Do you have an example of what such a non-modeled loss would be?

Agreed and modified: (e.g. a non-modeled peril such as storm surge induced by windstorms)

Line 116 Replace with "As more data.." for readability

Sentence has been removed as it does not bring additional information. Lines 102 to 104 explain how the modelling now includes information in hazard and vulnerability modules to capture not only extreme events but also smaller events.

Line 116, 117 It's not clear what is meant here. You need more data to look at extreme events, so the meaning is not clear. Can you rephrase?

Sentence has been removed as it does not bring additional information. Lines 102 to 104 explain how the modelling now includes information in hazard and vulnerability modules to capture not only extreme events but also smaller events.

Line 156 "was" should be "is"

Agreed and modified

Line 171 Should it be two or three components? Only two reasons are listed in the rest of the sentence.

Modified: The next section focuses on three of the loss modelling framework's components highlighting where (i) a thorough and systematic data collection needs to be put in place, and (ii) the loss modelling framework requires investment to upgrade it and tailor it to respond to insurers' business needs.

Line 179 Repeated lines Line

The repetition does not appear in this version

185-190 I'm not sure what is meant in this paragraph. What are these gaps?

This sentence has been removed. It referred to the gap mentioned in Section 2.2. the paragraph in Section 2.2 has been rephrased to give more precision:

Today's IT computation constraints make it necessary to downgrade the quality and sophistication of the researchers' modelling to obtain results within an acceptable period. This compromises the assessment that could be attained and engenders a precision gap between what research produces and the derivative data ultimately integrated in the loss modelling framework.

Line 214 Repeated lines

The repetition does not appear in this version

Line 216 What is the nature of these sensitivity tests?

These sensitivity tests consist in changing the physical properties of the building and assess the impact on the losses.

Sentence modified:

Any omission on the properties of a building's construction induces an uncertainty on that given building(s)'s exposure that can be quantified through sensitivity tests that assess varying combinations of a building's construction properties and the resulting impact on losses

Line 227 Repeated lines

The repetition does not appear in this version

Line 250 "ultimate" final would be a better word

Agreed and modified

Line 254 "rupture" damage would be a better word.

"Rupture" is kept, what is meant here is that the damage comes from a partly or complete rupture / breach of a building's components (windows, roof, walls…)

Line 272 Repeated lines

The repetition does not appear in this version

Line 289 Repeated lines

The repetition does not appear in this version

Line 296 "societal matter" I'm not sure what is meant by this – can you clarify. It seems to more about modelling and data handling in insurance than a matter for wider society.

If tackled collectively, data collection, especially in the area of damages and claims, could contribute to better city planning and more effective prevention measures, that would in turn increase society resilience.

Line 300. "modelling of loss modelling frameworks" should be plural as you're speaking more generally

It refers to the loss modelling framework, as defined in Section 2.2 and widely used in the (re)insurance market.

Line 315. I'm assuming your talking about ensemble modelling here? If not, can you be more specific what you are proposing for this "uncertainty component"

It is indeed the idea of ensemble modelling. It has been precised in the text:

How about introducing a specific "uncertainty component" that would deal with an ensemble of models combining multiple datasets from the different components and propagate the quantification all along the loss modelling process?

Section 4.3 It is important to explicitly use the words "climate change" in this section

Agreed and modified

---

## Author Response (AR3)

Dear Dr Peres,

Thank you very much for your comments and support all along the process.

Please find below the updated abstract:

This paper analyses how the current loss modelling framework that was developed in the 1990's to respond to hurricane Andrew market crisis falls short in dealing with today's complexity. In effect, beyond reflecting and supporting the current understanding and knowledge of risks, data and models are used in the assessment of situations that have not been experienced yet. To address this question, we considered the (re)insurance market's current body of knowledge on natural hazard loss modelling, the fruit of over 30 years' research conducted by (re)insurers, brokers, modelling firms, and other private companies and academics in the atmospheric sciences, geosciences, civil engineering studies, and data sciences among others. Our study shows that to successfully manage the complexity of the interactions between natural elements and the customer ecosystem, it is essential that both private companies in the insurance sector and academia continue working together to co-build and share a common data collection and modelling. This paper (i) proves the need to conduct an in-depth review of the existing loss modelling framework and (ii) makes it clear that only a transdisciplinary effort will be up to the challenge of building global loss models. These two factors are essential to capture the interactions and increasing complexity of the three risk drivers – exposure, hazard, and vulnerability – thus enabling insurers to anticipate and be equipped to face the far-ranging impacts of climate change and other natural events.

Minor changes have been done along the document and are highlighted in green in the version with track changes.